# Activity of Delafloxacin and Comparator Fluoroquinolones against Multidrug-Resistant *Pseudomonas aeruginosa* in an In Vitro Cystic Fibrosis Sputum Model

**DOI:** 10.3390/antibiotics12061078

**Published:** 2023-06-20

**Authors:** Vaughn D. Craddock, Evan L. Steere, Hannah Harman, Nicholas S. Britt

**Affiliations:** 1Department of Pharmacy Practice, University of Kansas School of Pharmacy, Lawrence, KS 66047, USA; 2Department of Population Health, University of Kansas School of Medicine, Kansas City, KS 66160, USA; 3Department of Internal Medicine, University of Kansas School of Medicine, Kansas City, KS 66160, USA

**Keywords:** delafloxacin, *Pseudomonas aeruginosa*, multidrug-resistance, carbapenem-resistance, carbapenem-resistant *Pseudomonas aeruginosa*, cystic fibrosis, fluoroquinolones

## Abstract

Delafloxacin (DLX) is a recently approved fluoroquinolone with broad activity against common cystic fibrosis (CF) pathogens, including multidrug-resistant *Pseudomonas aeruginosa* (MDR-Psa). Delafloxacin has been previously shown to have excellent lung and biofilm penetration and enhanced activity at lower pH environments, such as those that would be observed in the CF lung. We analyzed six Psa strains isolated from CF sputum and compared DLX to ciprofloxacin (CPX) and levofloxacin (LVX). Minimum inhibitory concentrations (MICs) were determined for DLX using standard culture media (pH 7.3) and artificial sputum media (ASM), a physiologic media recapitulating the CF lung microenvironment (pH 6.9). Delafloxacin activity was further compared to CPX and LVX in an in vitro CF sputum time-kill model at physiologically relevant drug concentrations (Cmax, Cmed, Cmin). Delafloxacin exhibited 2- to 4-fold MIC reductions in ASM, which corresponded with significant improvements in bacterial killing in the CF sputum time-kill model between DLX and LVX at Cmed (*p* = 0.033) and Cmin (*p* = 0.004). Compared to CPX, DLX demonstrated significantly greater killing at Cmin (*p* = 0.024). Overall, DLX demonstrated favorable in vitro activity compared to alternative fluoroquinolones against MDR-Psa. Delafloxacin may be considered as an option against MDR-Psa pulmonary infections in CF.

## 1. Introduction

Cystic fibrosis (CF) is a genetic disorder characterized by mutations in the cystic fibrosis transmembrane regulator (CFTR) and physiologic alterations in the lung [1,2]. The CFTR mutation leads to dysfunctional chloride channels in epithelial cells, resulting in abnormal respiratory secretions and inflammation [1]. Excessive mucus production obstructs respiratory tract cilia and impairs mucosal defense, ultimately leading to persistent infection [1,2]. *Pseudomonas aeruginosa* (Psa) is frequently implicated in CF lung disease and associated with multidrug resistance (MDR), decreased lung function, frequent hospitalizations, and significant mortality [3,4]. In order to manage chronic infections caused by Psa and other bacteria, CF patients are persistently prescribed antimicrobials to treat and prevent pulmonary exacerbations. Fluoroquinolones, such as ciprofloxacin (CPX) and levofloxacin (LVX), act as bacterial DNA synthesis inhibitors and are currently the only antimicrobials with reliable in vitro activity against Psa that can be administered orally. [3,4]. This is particularly advantageous in the management of CF due to the need for prolonged treatment and suppression of Psa without the requirement of central venous catheters. Additionally, recent data has demonstrated that CFTR modulators such as ivacaftor, which is commonly utilized in the treatment of CF, may synergize with fluoroquinolones against biofilm-producing Psa and enhance bacterial killing [5].

Unfortunately, resistance to CPX and LVX is common among Psa isolated from CF patients, necessitating alternative therapeutic strategies. In a recent study, nearly half of Psa isolates from CF sputum were resistant to CPX; however, many CPX-resistant Psa isolates retained susceptibility to a novel fluoroquinolone, delafloxacin (DLX), which has been recently approved for clinical use in the United States [6]. Delafloxacin has been previously shown to have excellent lung and biofilm penetration and features a unique chemical structure resulting in enhanced activity at lower pH environments, such as those that would be observed in the CF lung [7,8,9,10,11]. Multiple studies have reported decreased airway mucus pH in CF, particularly during pulmonary exacerbation [8,12,13,14,15]. Furthermore, Psa persists in a more acidic microenvironment in the biofilm phase compared to the fluid phase (5.6 versus 7.0), which is particularly relevant to CF lung disease where Psa often entrenches in biofilms [2,16]. Based on these properties, DLX may represent a novel therapeutic option uniquely suited for managing pulmonary infections in CF. However, there is a lack of data on the comparative activity of DLX and other fluoroquinolones against clinical Psa isolates from patients with CF and versus Psa biofilms.

Due to the unique properties of DLX, we hypothesized this agent would demonstrate enhanced activity compared to other commonly utilized fluoroquinolones. Therefore, the objective of this study was to evaluate the antibacterial activity of DLX and comparator fluoroquinolones against MDR-Psa isolated from CF sputum in an in vitro biofilm model simulating the microenvironment of the CF lung.

## 2. Results

### 2.1. Susceptibility Testing

To evaluate the comparative activities of fluoroquinolones against clinical Psa strains isolated from CF sputum, we determined the minimum inhibitory concentrations (MICs) for DLX and comparators. These results are summarized in Table 1. In cation-adjusted Mueller-Hinton broth (CAMHB), 4/6 strains were susceptible to DLX, 1/6 strains (B661) were intermediate, and 1/6 strains were resistant (B677) according to 2018 U.S. Food and Drug Administration (FDA) interpretation criteria [17]. Among the strains susceptible to DLX, all were susceptible to CPX and LVX, except for ATCC^®^ BAA-2108, which was LVX-intermediate according to Clinical Laboratory and Standards Institute (CLSI) breakpoints [18]. Delafloxacin-intermediate strain B661 was CPX-intermediate and LVX-resistant, and DLX-resistant strain B677 was CPX-intermediate and LVX-resistant.

To better determine the activity of DLX and comparators in a microenvironment more similar to the CF lung, we conducted MIC testing in artificial sputum media (ASM). As summarized in Table 1, DLX MICs were similar for 3/6 strains (range, 2- to 4-fold reduction) and unchanged for the other strains in ASM compared to DLX MICs in CAMHB. Delafloxacin exhibited a 4-fold MIC reduction against mucoid B660 and MICs were similar (within 1 dilution) against the DLX-intermediate strain (ATCC*^®^* BAA-2108). Ciprofloxacin MICs were increased 4-fold in 3/6 strains (B660, B310, and ATCC*^®^* BAA-2108) and similar (within 1 dilution) in the other 3/6 strains in ASM compared to CAMHB during planktonic growth. Levofloxacin MICs were increased 4-fold in 1/6 strains (B660), similar (within 1 dilution) in 2/6 strains (B310 and ATCC*^®^* BAA-2108) and remained unchanged in the other 3/6 strains using ASM compared to CAMHB. To determine whether these MIC shifts were due to pH changes or other ASM components, we performed MIC testing in pH-adjusted (pH, 6.9) CAMHB. Delafloxacin, CPX, and LVX MICs were similar in pH-adjusted CAMHB compared to ASM (Table 1).

### 2.2. Scanning Electron Microscopy (SEM)

To determine the utility of ASM for Psa biofilm formation, we performed SEM of bacterial cultures over a 72 h period. As shown in Figure 1A–D, Psa biofilm microcolony and early macrocolony formation were observed by SEM as early as 24 h after cultivation in ASM using strain ATCC^®^ BAA-2108. Maturing Psa biofilm formation was observed at 48 h (Figure 1B) and fully mature biofilms were observed at 72 h (Figure 1C). Similar biofilm characteristics were observed by SEM for clinical strains B660 (mucoid morphotype) and B310 (flat morphotype) following 72 h of cultivation in ASM. These images support the utility of ASM as a physiologic media for modeling antibacterial activity against Psa biofilms using both clinical strains and commercially available controls.

### 2.3. CF Sputum Biofim Time-Kill Model

The time-kill method was utilized to evaluate the pharmacodynamics of DLX and comparator fluoroquinolones over 24 h in a CF sputum (ASM) biofilm model. Concentrations tested in these experiments correspond to approximate physiologic maximal concentrations (Cmax), median concentrations (Cmed), and minimum concentrations (Cmin) which would be expected to be observed in vivo. This allows for a more comprehensive evaluation of antibacterial activity over a standard human dosing interval. Time-kill curves for DLX in the CF sputum model are displayed in Figure 2. As shown, DLX was rapidly bactericidal at all concentrations tested against DLX-susceptible isolates (B727, B660, B310, and ATCC^®^ BAA-2108), achieving over 99.9% bacterial killing from the average starting inoculum within 4 h (Figure 2A,B,E,F). Against the DLX-intermediate (B661; Figure 2C) and DLX-resistant (B677; Figure 2D) strains, DLX was bactericidal at concentrations of 4 and 10 µg/mL but did not have sustained activity at 1 µg/mL, as regrowth was observed (Figure 2C,D).

As depicted in the time-kill curves for CPX in the CF sputum model (Figure 3A–E), CPX achieved over 99.9% bacterial killing from the average starting inoculum at the simulated Cmax concentration of 4 µg/mL against all strains except ATCC*^®^* BAA-2108 (Figure 3F), against which CPX was bacteriostatic. At the simulated Cmed concentration of 2 µg/mL, CPX was bactericidal against 2/6 strains (B727 and B660, Figure 3A,B), bacteriostatic against 3/6 strains (B661, B677, and B310, Figure 3C–E), and inactive against 1/6 strains (ATCC*^®^* BAA-2108, Figure 3F). At the lowest CPX concentration of 1 µg/mL, bactericidal activity was achieved against only 1/6 strains (B727, Figure 3A) and activity was limited against all other strains (Figure 3B–F).

Time-kill curves for LVX in the CF sputum model are displayed in Figure 4. As depicted, LVX was bactericidal at simulated Cmax (10 µg/mL) against all 6 strains tested. Antibacterial activity was variable at simulated Cmed (4 µg/mL), with bactericidal activity observed against 2/6 strains (B727 [Figure 4A] and B310 [Figure 4E]), bacteriostatic activity against 2/6 strains (B660 [Figure 4B] and B677 [Figure 4D]), and inactivity against 2/6 strains (B661 [Figure 4C] and ATCC*^®^* BAA-2108 [Figure 4F]). At the simulated Cmin concentration (1 µg/mL), LVX activity was poor, with no bactericidal killing observed and bacteriostatic activity only achieved against 1/6 strains (B727, Figure 4A).

Comparing the 24 h antibacterial killing effects of simulated Cmax between DLX, CPX, and LVX in this model, no statistically significant differences were observed by analysis of variance (*p* = 0.578). However, there were significant differences in antibacterial activity between treatment groups at simulated Cmed (*p* = 0.033) and Cmin (*p* = 0.004) concentrations. Specifically, DLX had significantly increased killing effects compared to LVX at Cmed (adjusted *p* = 0.025) and Cmin (adjusted *p* = 0.005) concentrations and compared to CPX at Cmin (adjusted *p* = 0.024). There were no statistically significant differences between CPX and LVX at any of the concentrations tested.

## 3. Discussion

The principal finding of this study was that DLX demonstrated favorable in vitro activity against MDR-Psa in a CF sputum time-kill model recapitulating the conditions of the CF lung. Delafloxacin and CPX were similarly potent in standard culture media as compared to LVX. However, DLX demonstrated improved antibacterial activity compared to CPX and LVX in ASM and pH-adjusted CAMHB against some strains, supporting the potential for enhanced activity of this drug in vivo during the management of MDR-Psa infections in patients with CF and in lower pH microenvironments. Delafloxacin exhibited significantly better antibacterial activity than comparator fluoroquinolones in CF sputum time-kill experiments, particularly at lower simulated physiologic concentrations. In other words, DLX may be more likely to achieve sustained activity for a longer duration of the dosing interval compared to other fluoroquinolones against certain strains of MDR-Psa in CF.

It is notable that translating in vitro antibiotic susceptibility testing and antibacterial activity to clinical responses in CF patients is complex [19]. This is due to multiple factors, including CF lung microbiome dynamics and the propensity for Psa to entrench in biofilms. Biofilms complicate the treatment of pulmonary exacerbations caused by Psa because they hinder the access of antimicrobials to target cells, resulting in MIC shifts 10 to 1000 times that of planktonic growth [20,21]. Therefore, the determination of antibacterial activity against Psa grown in biofilms may provide a more accurate assessment of the utility of antimicrobial agents for the treatment of CF pulmonary exacerbations caused by Psa. Commonly utilized in vitro Psa biofilm models require cultivation on abiotic surfaces, such as polystyrene pegs or other polymers [22,23]. These models are useful for studying biofilms associated with medical device infections but are not representative of the mucus-embedded biofilms observed in CF airways. To mimic the unique microenvironment of the CF lung more closely, we used a physiologic media (ASM) to better simulate the growth conditions of Psa in the CF lung [24,25]. The present study features the novel use of ASM in an in vitro time-kill kinetic biofilm model and provides a framework for future evaluations of antibacterials against MDR-Psa and other pathogens of concern in CF. The correlation of antibacterial activities from this model to clinical response in CF pulmonary exacerbation is needed in follow-up studies.

We hypothesize that the improved activity of DLX against MDR-Psa in the CF sputum time-kill model is driven by biochemical changes of the drug in a lower pH microenvironment. The unique anionic structure of DLX, which exists predominantly in neutral form with decreasing pH, allows for enhanced bacterial uptake compared to alternative fluoroquinolones that carry a net positive charge at a lower pH [11]. In previous studies, DLX MICs decreased 4- to 7-fold when pH was lowered from 7.2 to 5.5, whereas MICs with comparator fluoroquinolones were increased or unchanged [7,11]. While the pH of the CF lung microenvironment is patient-dependent, the mean pH in submucosal gland fluid is 6.6–7.0 [26]. Lower pH measurements are reported in explanted CF airway tissue (pH range, 6.0–6.9), exhaled breath condensate in stable CF patients (mean pH, 5.88), and during pulmonary exacerbation (mean pH, 5.32) [27,28]. In the present study, the DLX MIC decreases observed in the CF sputum model and pH-adjusted CAMHB were modest and strain-specific. The significance of these decreases is of unclear clinical relevance, given the expected variation in MIC testing. Additional studies of DLX and fluoroquinolone activity in ASM at a lower pH may also be warranted. In the present study, time-kill assays were performed, which agreed with observed MIC values. However, caution should be taken when interpreting MIC data given known issues with the precision of MIC estimates.

This study has a number of limitations that should be considered. First, it included a limited number of clinical MDR-Psa isolates from CF patients experiencing pulmonary exacerbation at a single center in the United States. Ideally, our results should be reproduced in a broader sample of Psa phenotypes in patients with varying stages of CF lung disease. While the model we used included physiologically relevant levels of DLX and comparator fluoroquinolones, static concentrations were used for a limited time (24 h). Static drug concentrations do not simulate the dynamic pharmacokinetic (PK) exposures observed in vivo, which may influence antibacterial activity and resistance development [29]. Additionally, longer durations of antimicrobial therapy are typically utilized in the management of CF pulmonary exacerbations, but the time-kill model limits the time in which antibacterial effects can be evaluated. Unfortunately, there are currently no validated in vitro PK/pharmacodynamic (PD) biofilm models which simulate the CF lung microenvironment and allow for the evaluation of dynamic drug exposures over longer time intervals.

In summary, we report enhanced activity of DLX in comparison to alternative fluoroquinolones against some strains of MDR-Psa in an in vitro time-kill kinetic model recapitulating the microenvironment of the CF lung. Although further research is warranted, this study provides data supporting a potential role for DLX in the management of MDR-Psa pulmonary infections in patients with CF and limited treatment options.

## 4. Materials and Methods

### 4.1. Bacterial Strains

Five clinical Psa sputum isolates (B727, B660, B661, B677, B310) from 4 separate adult patients admitted for CF pulmonary exacerbation were collected at the University of Kansas Hospital (Kansas City, KS, USA). Strains B660 (mucoid) and B661 (flat) were isolated from the same patient during the same admission. Isolates were chosen from our collection to represent a spectrum of fluoroquinolone susceptibility phenotypes. ATCC^®^ BAA-2108 was used as a well-characterized commercially available carbapenem-resistant reference strain, previously isolated from CF sputum. ATCC^®^ 27,853 was used as the control strain for susceptibility testing [30].

### 4.2. Antimicrobials

Pharmaceutical grade DLX, CPX, and LVX powders were commercially purchased (Sigma-Aldrich, St. Louis, MO, USA). Fresh antimicrobial stocks were prepared daily according to CLSI recommendations [30].

### 4.3. Media

Cation-adjusted Mueller-Hinton broth (CAMHB; BD Difco, Detroit, MI, USA) was used as standard culture media for MIC testing, according to CLSI recommendations [30]. Artificial sputum media (ASM) was used as physiologic media for MIC testing and in time-kill models. This media contains amino acids, mucin, and exogenous DNA and was designed to recapitulate the microenvironment of the CF lung to allow for the formation of Psa biofilm aggregates similar to those observed in CF sputum [19,24,25,31]. All ASM components were commercially purchased, including DNA from fish sperm (Fisher Scientific, Waltham, MA, USA; catalog number 50-247-480), type II mucin from porcine stomach (Sigma-Aldrich, St. Louis, MO, USA; catalog number M2378), and individual amino acids (Fisher Scientific, Waltham, MA, USA; catalog numbers available upon request). For consistency, the same batches of each of the ASM components were utilized throughout all experiments. To ensure sterility, compounded ASM underwent pre-filtration with a 0.45-micron filter system followed by a second and final filtration in a 0.22-micron filter system. Following preparation, ASM was aliquoted and stored for up to 30 days refrigerated at 6 °C. Each experiment utilizing ASM also included a media control to ensure no contamination had occurred.

Because ASM is adjusted to pH 6.9 to mimic the acidic environment of the CF lung, pH-adjusted CAMHB was also used as a comparator. For pH-adjusted CAMHB, CAMHB was prepared using standard procedures, autoclaved, allowed to cool overnight at room temperature, and then adjusted to a pH of 6.9 using 0.1 M hydrochloric acid. After acidification, the media was sterilized using a 0.22-micron filter system. Mueller-Hinton agar (MHA; BD Difco, Detroit, MI, USA) was used for colony counts.

### 4.4. Susceptibility Testing

Minimum inhibitory concentration (MIC) testing was performed for DLX, CPX, and LVX by broth microdilution in CAMHB, ASM, and pH-adjusted CAMHB [20]. Susceptibilities were determined using 2018 U.S. Food and Drug Administration breakpoints for DLX and 2022 CLSI interpretive criteria for CPX and LVX [32,33].

### 4.5. Scanning Electron Microscopy

To visualize biofilm growth dynamics in ASM over time, bacterial cell cultures were visualized by SEM after 24 h, 48 h, and 72 h conditioning. At each time point, bacterial cells were fixed with 2% glutaraldehyde in 0.1 M cacodylate buffer for 24 h and then rinsed with 0.1 M cacodylate buffer. Samples were post-fixed in 1% osmium tetroxide in 1% cacodylate buffer for 30 min, rinsed 3 times with distilled water, and dehydrated using 50–100% ethanol (6 steps). Samples were further processed using bone dry grade CO_2_ in an EMS 850 critical point drier, mounted, and sputter coated with gold/palladium (EMS/Quorum 150 R ES Sputter Coater, Quorum Technologies, Lewes, UK). Cells were viewed using a Hitachi S-2700 SEM and images were captured with the Quartz PCI system.

### 4.6. CF Sputum Time-Kill Model

Delafloxacin time-kill models were performed using concentrations of 1, 4, and 10 µg/mL in ASM; CPX time-kill models were performed using concentrations of 1, 2, and 4 µg/mL in ASM; and LVX time-kill models were performed using concentrations of 1, 4, and 10 µg/mL in ASM. These concentrations were chosen to approximate respective minimum (Cmin), median (Cmed), and maximal (Cmax) concentrations observed in vivo during a standard dosing interval [17,32,33]. Bacterial suspensions were calibrated to a 0.5 MacFarland standard and diluted to yield a standard inoculum of approximately 5 × 10^5^ colony-forming units (CFU)/mL and drug solution was added to ASM. After all these components had been added, the final volume was 2 mL in each well. Cultures were incubated at 35 °C under aerobic conditions with continuous shaking (75 rpm). Pharmacodynamic samples were drawn at 0, 4, 8, and 24 h, serially diluted in cold normal saline, drop plated on MHA, and incubated for 24 h at 35 °C. Colonies were counted by visual inspection. All in vitro experiments were conducted in at least triplicate to ensure reproducible and precise pharmacodynamic estimates. Bactericidal activity was defined as a ≥3 log10 CFU/mL reduction in bacterial density relative to the starting inoculum. Bacteriostatic activity was defined as a 0–3 log10 CFU/mL reduction in bacterial density, and inactivity was defined as an increase (growth) in bacterial density relative to the starting inoculum.

### 4.7. Statistical Analysis

Antibacterial activity, as measured by log10 reduction in CFU/mL, was compared between simulated Cmax, Cmed, and Cmin of DLX, CPX, and LVX at 24 h in time-kill models using one-way analysis of variance (ANOVA) with Tukey’s post-hoc test. Each time-kill experiment was conducted in at least triplicate to ensure reproducible and precise estimates of 24 h killing. GraphPad Prism (version 9, GraphPad Software, La Jolla, CA, USA) was used for analysis and *p* ≤ 0.05 was considered statistically significant after adjustment for multiple comparisons.

## Figures and Tables

**Figure 1 antibiotics-12-01078-f001:**
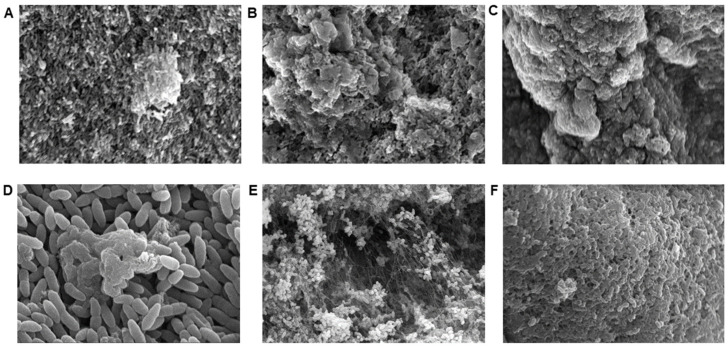
Biofilm growth dynamics of clinical and control strains were visualized in ASM by scanning electron microscopy (SEM). Biofilm growth dynamics of Psa ATCC^®^ BAA-2108 were observed by SEM at 2000× magnification after 24 h (**A**), 48 h (**B**), and 72 h (**C**) of cultivation in ASM. Early microcolony formation is observed at 10,000× magnification as early as 24 h of cultivation (**D**). Mature 72 h biofilms of clinical Psa strains B660 (**E**) observed by SEM at 2000× magnification and B310 (**F**) observed by SEM at 3000× magnification demonstrate the reproducibility of biofilm cultivation in ASM.

**Figure 2 antibiotics-12-01078-f002:**
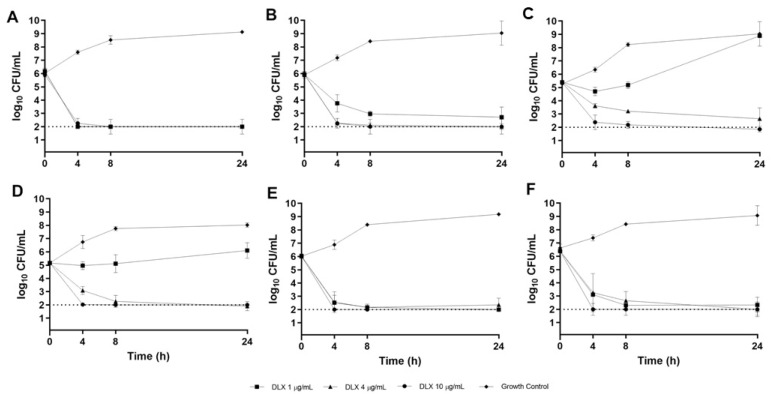
Delafloxacin (DLX) 24 h time-kill curves versus MDR-Psa CF sputum isolates B727 (**A**), B660 (**B**), B661 (**C**), B677 (**D**), B310 (**E**), and ATCC*^®^* BAA-2108 (**F**) in the CF sputum model. The average inoculum and standard deviation for each time point are depicted. Each experiment was repeated in at least triplicate. The dotted line indicates the theoretical lower limit of detection.

**Figure 3 antibiotics-12-01078-f003:**
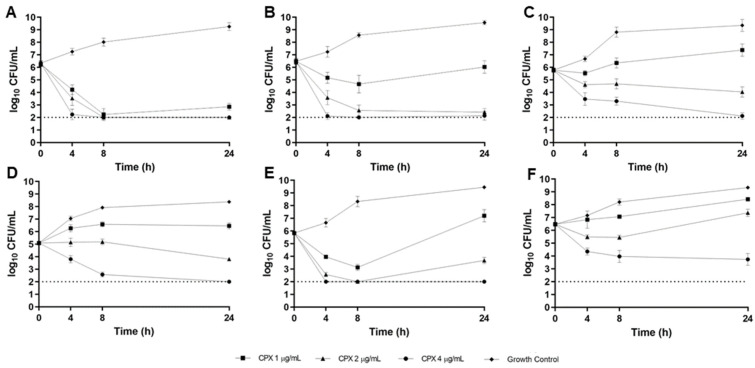
Ciprofloxacin (CPX) 24 h time-kill curves versus MDR-Psa CF sputum isolates B727 (**A**), B660 (**B**), B661 (**C**), B677 (**D**), B310 (**E**), and ATCC*^®^* BAA-2108 (**F**) in the CF sputum model. The average inoculum and standard deviation for each time point are depicted. Each experiment was repeated in at least triplicate. The dotted line indicates the theoretical lower limit of detection.

**Figure 4 antibiotics-12-01078-f004:**
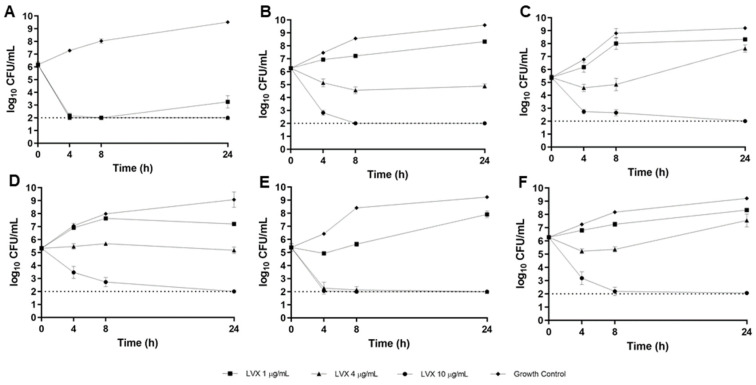
Levofloxacin (LVX) 24 h time-kill curves versus MDR-Psa CF sputum isolates B727 (**A**), B660 (**B**), B661 (**C**), B677 (**D**), B310 (**E**), and ATCC*^®^* BAA-2108 (**F**) in the CF sputum model. The average inoculum and standard deviation for each time point are depicted. Each experiment was repeated in at least triplicate. The dotted line indicates the theoretical lower limit of detection.

**Table 1 antibiotics-12-01078-t001:** Delafloxacin, ciprofloxacin, and levofloxacin minimum inhibitory concentrations versus clinical *Pseudomonas aeruginosa* cystic fibrosis sputum isolates in varying culture media.

	CAMHB (pH 7.3)			ASM (pH 6.9)			Adjusted CAMHB (pH 6.9)		
Isolate and Morphotype	DLX	CPX	LVX	DLX	CPX	LVX	DLX	CPX	LVX
B727 (flat)	0.0156	0.25	0.5	0.0156	0.5	0.5	0.0156	0.25	0.5
B660 (mucoid)	0.25	0.25	1	0.0625	1	4	0.125	1	4
B661 (flat)	1	1	8	1	2	8	0.5	1	8
B677 (flat)	2	1	4	1	2	4	1	1	4
B310 (flat)	0.25	0.25	1	0.25	1	2	0.25	0.5	1
ATCC^®^ BAA-2108 (flat)	0.5	0.5	2	0.25	2	4	0.5	1	4

All values are reported in µg/mL. CAMHB, cation-adjusted Mueller-Hinton broth; ASM, artificial sputum media; DLX, delafloxacin; CPX, ciprofloxacin; LVX, levofloxacin; MIC, minimum inhibitory concentration. Delafloxacin and CPX MIC breakpoints: susceptible, ≤0.5 µg/mL; intermediate, 1 µg/mL; resistant, ≥2 µg/mL. Levofloxacin MIC breakpoints: susceptible, ≤1 µg/mL; intermediate, 2 µg/mL; resistant, ≥4 µg/mL.

## Data Availability

The data presented in this study are available on request from the corresponding author.

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
