# Peer review of "Activity of Delafloxacin and Comparator Fluoroquinolones against Multidrug-Resistant Pseudomonas aeruginosa in an In Vitro Cystic Fibrosis Sputum Model"

_antibiotics, 2023, doi:10.3390/antibiotics12061078_

Round 1

Reviewer 1 Report

Comments are attached.

Author Response

1. The major flaw of the manuscript concerns the interpretation of differences between MIC values. The authors did not use any criteria, considering as “different” any difference. However, it is commonly accepted that two MICs are considered “significantly” if differ for ≥ 2-log2, i.e., 4-fold difference, because of variations occurring between different biological/technical replicates.

We agree with the reviewer that 2-fold MIC variations are typically expected/allowed in standard MIC testing experiments. The difference in activity with some strains is supported in time-kill experiments, although we do note in the Discussion that the effect is modest. Given the unclear clinical significance of our findings, we have added the following to the Discussion:

"In the present study, the DLX MIC decreases observed in the CF sputum model and pH-adjusted CAMHB were modest and would likely be further enhanced at lower pH environments. The significance of these decreases is of unclear clinical relevance, given expected variation in MIC testing. It is possible that the antibacterial activity of DLX reported represents a conservative estimate of the effect which would be observed in vivo during CF pulmonary exacerbation at lower pH environments. However, additional studies of DLX and fluoroquinolone activity in ASM at lower pH should be conducted to confirm this hypothesis."

2. For these reasons, both Results and Discussion sections must be substantially rephrased considering the above cited interpretative criteria. For example, the statements at Lines 63, 66, and 69-70 cannot be justified by the findings shown in Table 1; the same in the Discussion section at L143-144, L169, and L199. On the contrary, the sentence at L73-74 is right and results from previous studies (LL173-174) confirms that differences of 1-log2 are not significant but merely due to technical variations but higher differences are needed to justify a “different” antibacterial activity.

We agree with the reviewer that the significance of these differences can be questioned based on the MIC methodology. However, we disagree that the description of the data are inaccurate. These experiments were conducted in at least triplicate with reproducible findings which are also consistent with time-kill assays, supporting the fact these are true MICs. As above (comment 2), we have added additional context to these findings in the Discussion section.

3. Results from Table 1 indicate that acidic pH, and not ASM, affects antibiotic activity although for some strains only (B660 for DLX; B660 and ATCC for CPX; B660 for LVX). The authors should report and discuss
this finding.

We agree with this comment and have edited the following sentence in the Discussion section: 

"Delafloxacin and CPX were similarly potent in standard culture media as compared to LVX. However, DLX demonstrated improved antibacterial activity compared to CPX and LVX in ASM and pH-adjusted CAMHB, supporting the potential for enhanced activity of this drug in vivo during the management of MDR-Psa infections in patients with CF and in lower pH microenvironments."

4. What is the added value of performing SEM analysis on biofilm formation? The ASM model is not “novel” as frequently reported by the authors – and its suitability to allow biofilm formation has already been reported in the literature. In addition, pictures should have the same magnification to allow comparative evaluation at different time-points. For example: fig1B (24h) does not seem different from fig 1D (72h) contrarily to what is stated at L84-85.

The use of SEM was to demonstrate biofilm growth dynamics of Psa against clinical and commercially available strains. This time-lapse SEM is not well-described in the current literature. 

While we agree with the reviewer that ASM has been used in other experiments, application of ASM as a physiologic media in a time-kill methodology against clinical MDR-Psa from CF patients has not been performed, to our knowledge. However, given the reviewer has taken offense to this, we have removed the word "novel."

5. Time kill kinetics were performed on planktonic cells and not on preformed biofilm samples. Therefore, the authors must rephrase 2.3. “CF sputum biofilm time-kill model”.

This has been changed throughout.

6. For the same reason, also the tile must be changed to avoid misunderstandings

This has been changed.

7. Use the acronyms consistently throughout manuscript.

Acronyms have been checked and edited for consistency. 

8. L47: “the pH microenvironment of Psa”, confusing, please rephrase.

We have edited this sentence as follows to enhance clarity:

"Furthermore, Psa persists in a more acidic microenvironment in biofilm phase compared to fluid phase (5.6 versus 7.0), which is particularly relevant to CF lung disease where Psa often entrenches in biofilms [2, 15]. "

9.Table 1: figure legend should indicate the breakpoint values for each antibiotic.

As suggested, we have added the breakpoints to the Table legend

10. Figures 2-3-4: Add “Results are shown as mean values ± SD (or SEM?)” and indicate how many replicates were performed.

This information has been added to the figure legends.

11.L246: it seems that 2 mL were added to each well; please rephrase for the sake of clarity.

We have edited this sentence as follows: 

"Bacterial suspensions were calibrated to a 0.5 MacFarland standard and diluted to yield a standard inoculum of approximately 5 x 105 colony-forming units (CFU)/mL and drug solution was added to ASM. After all these components had been added, the final volume was 2 mL in each well. "

12. 4.7 Statistical analysis: indicate how many technical and biological replicates were carried out.
Each experiment was repeated in at least triplicate and including growth and media controls. This detail has been added to this Section and Figure legends.

Reviewer 2 Report

Introduction:

In this manuscript authors show the recently approved antibiotic Delafloxacin (DLX) has activity against a high biofilm forming pathogen associated with cystic fibrosis pulmonary infection, Pseudomonas aeruginosa. As DXL has previously shown enhanced activity at lower pH similar to the CF airways and anti-biofilm capabilities, the group studied the response of CF clinical P. aeruginosa isolates to DLX compared with other readily available fluroquinolone antibiotics. Recapitulating the CF lung microenvironments through the use of artificial sputum media, in vitro time-kill models were conducted at physiologically relevant concentrations to evaluate DLX-mediated killing. In this report they find a 2- to 4-fold MIC reduction occurs, which corresponded to enhanced bacterial killing particularly Cmin (P=0.024).  Work was not presented nor explained in a particularly clear way. However the results are clear on the efficacy of DLX as a new, highly potent fluoroquinolone against clinical MDR P. aeruginosa isolates. From this I have a few questions:

1.      When are you adding the antibiotics treatment to your ASM cultures?

2.      How are you ensuring sufficient homogenisation of antimicrobials through ASM media?

3.      Why was SEM not additionally used to validate time-kill assay results?

Major concerns:

Introduction-

·        No definition of Cmax, Cmedian, Cmin included anywhere in the paper, please include this.

·        No mention of the mucus being the major factor for colonisation, this is important as ASM is used as the model for this study so please include

·        No mention of fluroquinolones mechanism of action to determine why they are frequently prescribed to CF patients- please include

·        Needs to have a section at the end of the introduction outlining what is shown in this paper/hypothesis/what is being tested, please include this

Methods-

·        Section 4.2- Please state which type of ASM is used due to the variation in components between media types. Please also state how extra components such as mucin, eDNA and antibiotics etc. were added as no references provided for this either

Results-

·        Please clearly define which of the 6 strains are mucoid and non-mucoid at the beginning of the results section – maybe a table to highlight the two strains that came from the same patient as a frame of reference

·        Needs an introduction to each section to understand reasoning for experiments and why each are being used

·        Section 2.1- Please state what you are defining as ‘susceptible’ ‘intermediate’ and ‘resistant’

·        Section 2.2- need to outline definition for what was considered a biofilm e.g. how many cells clustered together

·        Section 2.2- why ATCC BAA-2108 was used for SEM? What about the other clinical strains. Need to include for representation from international strain-panel and clinical (mucoid and non-mucoid representative because of different phenotypes in regard to polysaccharide/biofilm production). No need to have two images representing 24hr time point, please remove one

·        Section 2.3- a graph to show statistical differences would be beneficial here

·        Some SEM images after treatment would add to the time-kill the data to make the inclusion of Section 2.2 more relevant

Figures-

·        A lot more information is required in the figure legends about how assay was conducted/outputs measured/statistics conducted

·        Figures 2-4 keys needs to be larger and reformat so that you have ABC next to each other, then DEF next to each other underneath. Rearrange which strains are allocated ‘ABCDEF’ by which strains are susceptible to DLX and discussed first- potentially from table suggested in section 2.1 for which strains are mucoid you could designate shorthand numbers/letters to make referencing to each strain easier when looking at data

Minor issues:

·        Section 2.1- need to look at the abbreviations for the media- go back and the first mention say the full name and abbreviate in brackets e.g. cation-adjusted Mueller-Hinton broth (CAMHB) then use CAMHB from then on.

·        Section 2.3- need to state ‘Figure X(letter)’ near to where each strain is mentioned in the text for figures 2-4

·        Section 2.3- how is 99.9% calculated? State starting concentration and how this was calculated in the figure legends

·        Section 2.3- what statistical test was used to gain p-values? This is stated in the methods but needs to be mentioned in the text

Editorial:

247- please edit to 35 °C

Author Response

1. When are you adding the antibiotics treatment to your ASM cultures?

Antimicrobials were added to ASM at T0, as currently detailed in the Methods.

2. How are you ensuring sufficient homogenisation of antimicrobials through ASM media?

Cultures were incubated under continuous shaking at 75 rpm to ensure homogenization, as included in the Methods.

3. Why was SEM not additionally used to validate time-kill assay results?

SEM was not used to validate time-kill assays, because it is unable to differentiate viable from non-viable cells. SEM imaging was performed to demonstrate qualitative biofilm formation over time in the ASM. The gold standard method of CFU/mL reduction was used in time-kills to determine antibacterial activity. Other methods, such as confocal of fluorescence  microscopy may be useful but were beyond the scope of this project.

4. No definition of Cmax, Cmedian, Cmin included anywhere in the paper, please include this.

These refer to maximal, median, and minimum concentrations which are physiologically observed. This has been more clearly defined in the Methods section.

5. No mention of the mucus being the major factor for colonisation, this is important as ASM is used as the model for this study so please include

W agree with the reviewer and have added the introduction to better emphasize this point. The first 3 sentences now read as follows:

"Cystic fibrosis (CF) is a genetic disorder characterized by mutations in the cystic fibrosis transmembrane regulator (CFTR) and physiologic alterations in the lung [1, 2]. The CFTR mutation leads to dysfunctional chloride channels in epithelial cells, resulting in abnormal respiratory secretions and inflammation [1]. Excessive mucus production obstructs respiratory tract cilia and impairs mucosal defense, ultimately leading to persistent infection [1, 2]."

6. No mention of fluroquinolones mechanism of action to determine why they are frequently prescribed to CF patients- please include

The mechanism of action of fluoroquinolones (DNA synthesis inhibitor) has been added to the Introduction. The chemical structure and mechanism of enhanced activity under acidic pH is currently discussed in the Introduction and Discussion sections. The availability of fluoroquinolones as oral agents is particularly advantageous in CF as patients requires prolonged courses of therapy. Additionally, there are recent data highlighting synergy between CFTR modulators and fluoroquinolones, which may enhance the appeal of these agents in CF. The following has been added to the Introduction section:

"Fluoroquinolones, such as ciprofloxacin (CPX) and levofloxacin (LVX), act as bacterial DNA synthesis inhibitors and are currently the only antimicrobials with reliable in vitro activity against Psa that can be administered orally. [3, 4]. This is particularly advantageous in the management of CF due to the need for prolonged treatment and suppression of Psa without requirement of central venous catheters. Additionally, recent data has demonstrated that CFTR modulators such as ivacaftor, which is commonly utilized in the treatment of CF, may synergize with fluoroquinolones against biofilm-producing Psa and enhance bacterial killing." 

7. Needs to have a section at the end of the introduction outlining what is shown in this paper/hypothesis/what is being tested, please include this

The objective of the study is currently described in the last sentence of the Introduction. We have added the following sentence to add our hypothesis and separated these last 2 sentences in a separate paragraph to add emphasis for the reader:

"Due to the unique properties of DLX, we hypothesized this agent would demonstrate enhanced activity compared to other commonly utilized fluoroquinolones." 

8. Please state which type of ASM is used due to the variation in components between media types. Please also state how extra components such as mucin, eDNA and antibiotics etc. were added as no references provided for this either

The methods in the citations were used to generate the ASM. No other antimicrobials other than those that were tested (delafloxacin, levofloxacin, ciprofloxacin) are added to the media. To add clarity, we have added manufacturer/supplier information for the mucin and DNA components as part of the manuscript. Additionally we have added further details regarding ASM preparation. The following has been added:

"All ASM components were commercially purchased, including DNA from fish sperm (Fisher Scientific, Waltham, Massachusetts, USA; catalog number 50-247-480), type II mucin from porcine stomach (Sigma-Aldrich, St. Louis, Missouri, USA; catalog number M2378), and individual amino acids (Fisher Scientific, Waltham, Massachusetts, USA; catalog numbers available upon request). For consistency, the same batches of each of the ASM components were utilized throughout all experiments. To ensure sterility, compounded ASM underwent pre-filtration with a 0.45-micron filter system followed by a second and final filtration in a 0.22-micron filter system. Following preparation, ASM was aliquoted and stored for up to 30 days refrigerated at 6 °C. Each experiment utilizing ASM also included a media control to ensure no contamination had occurred.

Because ASM is adjusted to pH 6.9 to mimic the acidic environment of the CF lung, pH-adjusted CAMHB was also used as a comparator. For pH-adjusted CAMHB, CAMHB was prepared using standard procedures, autoclaved, allowed to cool overnight at room temperature, and then adjusted to a pH of 6.9 using 0.1M hydrochloric acid. After acidification, the media was sterilized using a 0.22-micron filter system." 

9. Please clearly define which of the 6 strains are mucoid and non-mucoid at the beginning of the results section – maybe a table to highlight the two strains that came from the same patient as a frame of reference

We have added the morphotype in parentheses next to each strain as part of Table 1.

10. Needs an introduction to each section to understand reasoning for experiments and why each are being used

We have added 1-2 sentences to each subsection of the Results to better explain the rationale for the set of experiments. We have also expanded upon the Methods section by adding more experimental detail.

11. Section 2.1- Please state what you are defining as ‘susceptible’ ‘intermediate’ and ‘resistant’

The definitions used are those that are defined by CLSI and US FDA for delafloxacin (no CLSI breakpoint currently). We have better emphasized this in Section 2.1 and have added citations. These are also mentioned and cited in the Methods section.

12. Section 2.2- need to outline definition for what was considered a biofilm e.g. how many cells clustered together

Unfortunately, there is not a uniform quantitative definition for this. The SEM images clearly depict commonly described biofilm structures seen with Pseudomonas aeruginosa, which we believe supports our use of the "biofilm" term throughout the manuscript.

13. Section 2.2- why ATCC BAA-2108 was used for SEM? What about the other clinical strains. Need to include for representation from international strain-panel and clinical (mucoid and non-mucoid representative because of different phenotypes in regard to polysaccharide/biofilm production). No need to have two images representing 24hr time point, please remove one

We chose to use this strain because it represents a commercially available carbapenem-resistant CF strain allowing future investigators to replicate or extend upon our findings. Unfortunately, we do not have images over time (24, 48, 72 h) for all clinical strains, but we have added 72 h (mature) biofilm images for other strains from this study which we do have on file (Figure 1E-F). This includes a clinical mucoid strain (B660). Due to cost/budget limitations, we were unable to perform SEM on all strains for this study. However, results were consistent across the strains we did test.

The following has been added to the Results section 2.2.:

"Similar biofilm characteristics were observed by SEM for clinical strains B660 (mucoid) and B310 (flat) following 72 h cultivation in ASM. These images support the utility of ASM as a physiologic media for modelling antibacterial activity against Psa biofilms using both clinical strains and commercially available controls."

14. Section 2.3- a graph to show statistical differences would be beneficial here

These statistical differences correspond to the time-kill curves depicted in Figures 2-4. We have added additional description in order to enhance clarity.

15. Some SEM images after treatment would add to the time-kill the data to make the inclusion of Section 2.2 more relevant

As above, we do not believe these images are relevant as they would not be able to discern cell viability.

16. A lot more information is required in the figure legends about how assay was conducted/outputs measured/statistics conducted

We agree with the reviewer that it may be helpful to the reader to have additional experimental detail included in the Figure legends. We have added additional information/methods to each of the Figure legends.

17. Figures 2-4 keys needs to be larger and reformat so that you have ABC next to each other, then DEF next to each other underneath. Rearrange which strains are allocated ‘ABCDEF’ by which strains are susceptible to DLX and discussed first- potentially from table suggested in section 2.1 for which strains are mucoid you could designate shorthand numbers/letters to make referencing to each strain easier when looking at data

We have rearranged these figures as suggested.

18. need to look at the abbreviations for the media- go back and the first mention say the full name and abbreviate in brackets e.g. cation-adjusted Mueller-Hinton broth (CAMHB) then use CAMHB from then on.

This has been corrected.

19. Section 2.3- need to state ‘Figure X(letter)’ near to where each strain is mentioned in the text for figures 2-4

We have added this additional detail throughout.

20. Section 2.3- how is 99.9% calculated? State starting concentration and how this was calculated in the figure legends

This is calculated from the average starting inoculum. This detail has been added to Section 2.3. We have added additional detail to the figure legends, including this information, as suggested in Comment 16.

21. Section 2.3- what statistical test was used to gain p-values? This is stated in the methods but needs to be mentioned in the text

We utilized ANOVA to calculate p-values. This has been added to the Results.

22. please edit to 35 °C

This has been corrected.

Reviewer 3 Report

In this study the authors evaluated the antibacterial activity of delafloxacin versus two fluoroquinolones comparator (ciprofloxacin and levofloxacin) against MDR-P. aeruginosa isolated from Cystic Fibrosis sputum in an in vitro biofilm model. Of interest is the novel use of artificial sputum media for MIC testing and in vitro time-kill kinetic biofilm model, simulating the microenvironment of the CF lung. Although the study includes very limited number of clinical isolates, the results seem to be encouraging and they were adequately explained and discussed. The study limitations were also well exposed.

There is just one thing that is not clear to me: in Table 1C the image looks very different from the others, as if it were in a different scale. Can the authors explain why?

The following corrections should be made in the text:

From lane 30 to 203: please distribute the test equally among the margins of the first 3 paragraphs (Justified Text).

Lane 57: please add the abbreviation MICs after the Minimun Inhibtory Concentration

Lane 75: in table 1 please make the font style equal and correct the last line, ATCC® BAA-2108 should be on one single line

Lane 247 and 249: please report correctly the temperature 35°C

Author Response

Thank you for your review and suggestions to improve this manuscript. Please see the following point-by-point response to the critiques:

There is just one thing that is not clear to me: in Table 1C the image looks very different from the others, as if it were in a different scale. Can the authors explain why?

Thank you for pointing this out. Images corresponding to Figure 1A, 1B, and 1D were taken at 2,000x magnification. The image in Figure 1C is a higher magnification (10,000x) image demonstrating the early macrocolony structures observed at 24 h. We have added this information to the caption to better inform the reader.

The following corrections should be made in the text:

From lane 30 to 203: please distribute the test equally among the margins of the first 3 paragraphs (Justified Text).

These sections of text have been justified.

Lane 57: please add the abbreviation MICs after the Minimun Inhibtory Concentration

Abbreviation added to line 57.

Lane 75: in table 1 please make the font style equal and correct the last line, ATCC® BAA-2108 should be on one single line

Have adjusted this formatting.

Lane 247 and 249: please report correctly the temperature 35°C

This correction has been made.

Round 2

Reviewer 1 Report

Dear Authors,

I'm still convinced that comments 1 and 2 must be addressed to overcome the main flaw of the manuscript, which is a wrong interpretation of differences in MIC values. 

Author Response

To better address the Reviewer's concern, we have removed the term "2-fold reduction" and replaced it with "similar (within 1 dilution)." The referenced paragraph in the Results section now reads as follows (lines 80-93, changed bolded):

"To better determine the activity of DLX and comparators in a microenvironment more similar to the CF lung, we conducted MIC testing in artificial sputum media (ASM). As summarized in Table 1, DLX MICs were similar for 3/6 strains (range, 2- to 4-fold reduction) and unchanged for the other strains in ASM compared to DLX MICs in CAMHB. Delafloxacin exhibited a 4-fold MIC reduction against mucoid B660 and MICs were similar (within 1 dilution) against the DLX-intermediate strain (ATCC® BAA-2108). Ciprofloxacin MICs were increased 4-fold in 3/6 strains (B660, B310, and ATCC® BAA-2108) and similar (within 1 dilution) in the other 3/6 strains in ASM compared to CAMHB during planktonic growth. Levofloxacin MICs were increased 4-fold in 1/6 strains (B660), similar (within 1 dilution) in 2/6 strains (B310 and ATCC® BAA-2108) and remained unchanged in the other 3/6 strains using ASM compared to CAMHB. To determine whether these MIC shifts were due to pH changes or other ASM components, we performed MIC testing in pH-adjusted (pH, 6.9) CAMHB. Delafloxacin, CPX, and LVX MICs were similar in pH adjusted CAMHB compared to ASM (Table 1)."

We chose to use the term "similar (within 1 dilution)" instead of "unchanged" to avoid confusion by the reader since the values listed are still numerically different and not technically unchanged. We acknowledge the lack of statistical significance of these differences given the limitations of the MIC testing methodology. This is mentioned in the Discussion section of the current version of the manuscript, in the following passage:

"The significance of these decreases is of unclear clinical relevance, given expected variation in MIC testing." (lines 220-221).

We hope these modifications have sufficiently addressed the Reviewer's concern.

Reviewer 2 Report

All the comments have been adressed.

Author Response

Thank you for your comment. We are pleased that all concerns were addressed.

Round 3

Reviewer 1 Report

I did not see any changes in the Discussion and Conclusions. 

Considering the new criteria for interpreting differences in MIC values has no impact on the sections mentioned above?

Author Response

In addition to the previous revisions to the Discussion section, we have made the following changes (in bold) to soften the language:

"However, DLX demonstrated improved antibacterial activity compared to CPX and LVX in ASM and pH-adjusted CAMHB against some strains.." (line 183)

"In other words, DLX may be more likely to achieve sustained activity for a longer duration of the dosing interval compared to other fluoroquinolones against certain strains of MDR-Psa in CF." (lines 188-190)

"In the present study, the DLX MIC decreases observed in the CF sputum model and pH-adjusted CAMHB were modest and strain-specific." (lines 219-221)

"Additional studies of DLX and fluoroquinolone activity in ASM at lower pH may be warranted." (lines 224-225)

The following sentence has been removed:

"It is possible that the antibacterial activity of DLX reported represents a conservative estimate of the effect which would be observed in vivo during CF pulmonary exacerbation at lower pH environments." (lines 222-224)

This manuscript does not include a Conclusion section.